# Development of a Machine Learning Model of Postoperative Acute Kidney Injury Using Non-Invasive Time-Sensitive Intraoperative Predictors

**DOI:** 10.3390/bioengineering10080932

**Published:** 2023-08-05

**Authors:** Siavash Zamirpour, Alan E. Hubbard, Jean Feng, Atul J. Butte, Romain Pirracchio, Andrew Bishara

**Affiliations:** 1School of Medicine, University of California, San Francisco, CA 94143, USA; 2Division of Biostatistics, School of Public Health, University of California, Berkeley, CA 94704, USA; 3Department of Epidemiology and Biostatistics, University of California, San Francisco, CA 94158, USA; 4Bakar Computational Health Sciences Institute, University of California, San Francisco, CA 94143, USAandrew.bishara@ucsf.edu (A.B.); 5Department of Anesthesia and Perioperative Care, University of California, San Francisco, CA 94143, USA

**Keywords:** acute kidney injury, artificial intelligence, clinical decision support, hemodynamic parameters, intraoperative predictors, machine learning, non-invasive monitoring, surgical complications, time-sensitive indicators

## Abstract

Acute kidney injury (AKI) is a major postoperative complication that lacks established intraoperative predictors. Our objective was to develop a prediction model using preoperative and high-frequency intraoperative data for postoperative AKI. In this retrospective cohort study, we evaluated 77,428 operative cases at a single academic center between 2016 and 2022. A total of 11,212 cases with serum creatinine (sCr) data were included in the analysis. Then, 8519 cases were randomly assigned to the training set and the remainder to the validation set. Fourteen preoperative and twenty intraoperative variables were evaluated using elastic net followed by hierarchical group least absolute shrinkage and selection operator (LASSO) regression. The training set was 56% male and had a median [IQR] age of 62 (51–72) and a 6% AKI rate. Retained model variables were preoperative sCr values, the number of minutes meeting cutoffs for urine output, heart rate, perfusion index intraoperatively, and the total estimated blood loss. The area under the receiver operator characteristic curve was 0.81 (95% CI, 0.77–0.85). At a score threshold of 0.767, specificity was 77% and sensitivity was 74%. A web application that calculates the model score is available online. Our findings demonstrate the utility of intraoperative time series data for prediction problems, including a new potential use of the perfusion index. Further research is needed to evaluate the model in clinical settings.

## 1. Introduction

Acute kidney injury (AKI) is a major postoperative complication associated with a higher risk of death, chronic kidney injury, long-term major adverse cardiovascular events, and cost of care [1,2,3,4,5,6,7,8,9,10]. Implementing reno-protective strategies has been shown to prevent episodes of AKI in high-risk patients identified by biomarkers [11]; however, lab tests for biomarkers are expensive and not widely available or used. Early identification of high-risk patients preoperatively using available data in the electronic health record (EHR) can be used to accurately stratify patients at risk of perioperative AKI and is highly likely to prevent episodes of AKI and improve outcomes by implementing early reno-protective strategies [12]. Intraoperatively, both the patient status and risk for potential complications rapidly evolve; hence, progression of risk factors can occur rapidly, and real-time analysis could provide timely instruction to aid an anesthesia provider.

Observational studies in noncardiac surgeries have found a 9.0–10.1% incidence of postoperative AKI [12,13]. In a randomized trial with careful postoperative monitoring, 12.3–13.4% developed AKI among the intervention groups; no intervention (aspirin, clonidine, or both) significantly reduced the risk of postoperative AKI [14].

Postoperative AKI often has a prerenal etiology whereby hypoperfusion causes an ischemic insult to tubular epithelial cells; in severe cases, epithelial cell death by apoptosis and necrosis manifests as acute tubular necrosis [15,16]. Thus, proposed intraoperative predictors of AKI include arterial and noninvasive blood pressures, arterial pressure variability, and urine oxygen partial pressure [13,17,18]. An observational study of noncardiac procedures in which 9.0% of patients developed AKI found no association between intraoperative hypotension and AKI in those with low preoperative risk, an association between severe intraoperative hypotension (mean arterial pressure < 50 mmHg) and AKI in participants with medium preoperative risk (adjusted odds ratio, 2.62), and an association between mild intraoperative hypotension (mean arterial pressure between 55 and 59 mmHg) and AKI in participants with high preoperative risk (adjusted odds ratio, 1.34) [13].

While postoperative AKI is often initiated by renal hypoperfusion, systolic blood pressure varies among arterial segments due to pulse pressure amplification; therefore, brachial or radial systolic blood pressure may not reliably measure renal hypoperfusion [19]. In a cohort study of 82,659 participants undergoing noncardiac surgery, arterial pressure standard deviation, coefficient of variation, variation independent of mean, and average real variability were associated with postoperative AKI independent of intraoperative hypotension, with adjusted standardized odds ratios of 1.11–1.14 per standard deviation [17]. In a prospective cohort study of 91 participants undergoing cardiac surgery, the adjusted relative risk of AKI was 0.82 per 10 mm Hg increase in mean urinary oxygen partial pressure [18]. Challenges of these approaches include the intermittent nature of noninvasive blood pressure monitoring, invasiveness of arterial pressure monitoring, and specialized equipment required for urinary oxygen monitoring. In a study of 42,615 major noncardiac surgeries, the addition of intraoperative variables to prehospitalization and preoperative variables in a predictive model of postoperative AKI resulted in a modest but significant increase in predictive performance (area under the receiver operating characteristic (ROC) curve (AUC), 0.82 vs. 0.80) [12]. Notably, this model represented intraoperative data using summary rather than time-sensitive measures and only evaluated indicators of central hypotension.

The objective of this study was to characterize non-invasive, time-sensitive intraoperative predictors of AKI. Our hypothesis was that a limited set of physiologically relevant intraoperative variables provides adequate prediction of postoperative AKI.

## 2. Materials and Methods

We conducted a retrospective cohort study from 2016 to 2022 at the University of California, San Francisco, CA, USA, an urban quaternary academic medical center. Inclusion criteria included adult operative cases during the study period with ≥1 serum creatinine (sCr) value in the 90 days preceding surgery and ≥1 serum sCr in the 48 h following surgery (Figure 1). Exclusion criteria included obstetric, kidney donor and recipient, and arteriovenous fistula cases due to preexisting alteration in renal physiology. For the same reason, we excluded those with last preoperative sCr ≥ 4.5 mg/dL. The study size was not prespecified.

This study was approved by the University of California, San Francisco Institutional Review Board (#17-23204) with a waiver of informed consent. STROBE guidelines for observational studies and TRIPOD guidelines for prediction models were followed [20,21].

Candidate preoperative and intraoperative predictors were selected based on being routinely measured, noninvasive, or suspected in the literature or the investigators’ clinical experience (Appendix A). A surgery-specific risk score was calculated as described previously [22]. Intraoperative variables were recorded at a frequency of 1/60 Hz. Non-invasive blood pressure and urine output were measured intermittently. Non-invasive blood pressure values were linearly interpolated. Urine output was back-calculated as a constant rate to the preceding urine output recording or the start of the case.

Intraoperative variables were investigated in a time-sensitive manner. The variables recorded at each minute of the operation included vital signs and signals from the pulse oximeter. Quantiles were determined for each variable at 2.5%, 5%, 10%, 25%, 50%, 75%, 90%, 95%, and 97.5% (Appendix A). To prevent longer cases from being overrepresented in these distributions, 100 values of each variable were resampled with replacement from each case prior to calculating the quantiles. We then calculated the number of minutes in each case during which the respective intraoperative variable was less than or equal to and above each respective quantile.

The main outcome was AKI according to the Kidney Disease Improving Global Outcomes (KDIGO) criteria in the 48 h following surgery [23]: [48 h maximum postoperative sCr] − [last preoperative sCr] ≥ 0.3 mg/dL or [48 h maximum postoperative sCr]/[last preoperative sCr] ≥ 1.5. While we attempted to incorporate KDIGO urine output criteria into our definition of postoperative AKI, we found that this was not well-suited to retrospective analysis due to unclear or inconsistent urine output charting practices, as was similarly found in other studies [12]. We restricted postoperative follow-up to 48 h rather than 7 days to avoid misclassifying cases in which AKI was more directly related to features of the postoperative rather than intraoperative course. Whereas prior studies have modeled AKI as a binary (stage 1 vs. stage 0) or multiclass (stages 0–3) outcome, we modeled the continuous difference between 48 h maximum postoperative sCr and last preoperative sCr, as we found this improved model performance. Cases with a postoperative sCr decrease were labeled as having a postoperative sCr increase of 0 for the purposes of modeling.

For variables with missing data, we added a binary indicator of missingness. All categorical variables were encoded by dummy variables. Missing data were imputed as the median for continuous variables and the most common category for categorical variables. Data were abstracted using Opal, an implementation science tool for clinical decision support in anesthesia [24].

All analyses were carried out in R version 4.1.1. Characteristics of the study population were summarized as the median and interquartile range (IQR) for continuous variables and counts and percentages for categorical variables. Differences between cases with and without AKI were summarized as the standardized mean difference.

Cases were randomly assigned to an 80% training or 20% internal validation set. Due to the large number of candidate intraoperative predictors, we carried out a two-step modeling strategy on the training set. To select main terms, we conducted 1000 trials of elastic net regression using the glmnet package [25]. Prior to commencing the trials, the alpha parameter was tuned to α = 0.5 among {0.000, 0.001, 0.008, 0.027, 0.064, 0.125, 0.216, 0.343, 0.512, 0.729, and 1.000} by minimizing the cross-validated loss value initialized using cva.glmet in the glmnetUtils package [26]. In each trial, cv.glmnet was called with nfolds = 5, alpha = 0.5, standardize = TRUE, and otherwise default arguments, to identify the largest regularization parameter lambda within one standard error of the lambda that minimized the cross-validated error (λ_1se_). We conducted repeated trials as suggested by developers of the glmnet package due to the inherent randomness of the folds.

We then assessed interactions among the main terms identified in the above procedure using the glinternet package [27]. We called glinternet.cv with nFolds = 5 and otherwise default arguments to identify λ_1se_. Of note, glinternet initially standardizes continuous features to unit norm and mean zero by default. We used this two-step strategy because of the computational infeasibility of assessing the >25,000 interactions in the initial set of candidate predictors. The glinternet model where λ = λ_1se_ was deemed the POSTOP-AKI (perfusion optimized score to predict AKI) model and evaluated in the validation set. Appendix A shows a flowchart of the variable selection and model building process. For comparison, the POSTOP-AKI model was compared to a simple preoperative predictor—the minimum sCr in the 90 days preceding the procedure.

We created ROC and precision-recall (PR) curves and calculated their respective AUCs using the continuous predicted postoperative increase in sCr. We conduced decision curve analysis using the dcurves package [28]. Because decision curve analysis necessitates that markers be bound within (0, 1), the minimum preoperative creatinine was transformed as such by logistic regression; the POSTOP-AKI scores already met this condition. We chose three thresholds for the predicted postoperative increase in sCr to facilitate analyses requiring a binary outcome. The middle threshold of 0.0767 was identified using the Youden index. Because interventions for AKI are relatively low risk, we also demonstrate a lower threshold of 0.05, which has higher sensitivity. The variance in the postoperative increase in sCr in the training set is 0.05. We also selected a higher threshold of 0.1, which has a higher positive predictive value, to provide users with scenarios wherein higher risk interventions may be warranted. These thresholds were meant to be demonstrative and should not be interpreted as optimized in external data sets. We created confusion matrices and calculated specificity, sensitivity, and positive predictive value (PPV) using these three cutoffs. A calibration curve was created by plotting the observed values for postoperative increase in sCr against predicted values in the validation set. *R*^2^ and the root-mean-square error (RMSE) were calculated for the calibration curve in the validation set. Moreover, the observed rate of AKI as a binary outcome was plotted within deciles of the predicted postoperative increase in sCr.

We then assessed predictive performance in subpopulations of the validation set defined by cutoffs in the 90-day minimum preoperative sCr due to the influence of preoperative kidney function on the risk of developing postoperative AKI. Across 200 equally spaced cutoffs from the 10th percentile (0.49) to the 90th percentile (1.46) of minimum preoperative sCr, we calculated ROC AUCs in the subpopulation of the validation set with minimum preoperative sCr greater than or equal to the cutoff.

We then compared random forest and XGBoost models to the POSTOP-AKI model using the caret, ranger, and xgboost packages [29,30,31]. For hyperparameter optimization, we called the train function on the same training set as above with method = ‘ranger’ and ‘xgbTree’, metric = ‘RMSE,’ and the trainControl parameters method = ‘cv’, number = 10, and search = ‘grid’. The predictors were those identified by the elastic net variable selection step described above (2 preoperative variables and 4 intraoperative variables). We created ROC and PR curves and calculated their respective AUCs as described above.

To aid interpretation of the variables included in the POSTOP-AKI model, we conducted OLS regression on 1000 bootstraps of the training set using the main terms identified above. Confidence intervals were calculated by the percentile method using the boot package [32], though they should not be interpreted in the context of statistical hypothesis testing due to the numerous variable selection steps in our model-building approach. Linear relationships were evaluated by plotting the observed rate of postoperative AKI as a function of the number of minutes meeting intraoperative thresholds in 30 min bins, the estimated blood loss in 100 mL bins, and the minimum preoperative sCr in 0.2-mg/dL bins.

Trends in retained model variables were further explored by plotting the proportion of non-AKI and AKI cases meeting predictive thresholds for urine output, heart rate, and perfusion index at each minute during the procedure.

## 3. Results

### 3.1. Patient Characteristics

We initially evaluated 77,428 adult operative cases not involving obstetric, kidney transplant, or AV fistula surgery during the study period (Figure 1). A total of 11,212 cases were further evaluated based on existing sCr data in the 90 days preceding and 48 h following the procedure. A total of 589 cases with preoperative sCr ≥ 4.5 mg/dL were excluded from the analysis, resulting in an analytic set of 10,623 cases. Then, 8519 were randomly assigned to the training set and 2104 to the validation set. There were 469 (5.5%) and 132 (6.3%) cases with AKI in the training and validation sets, respectively.

Patients had a median [IQR] age of 62 (51, 71) years. A total of 5871 (55.3%) patients were male and 6401 (60.3%) identified as White or Caucasian. A total of 2675 (25.2%) patients were classified as an ASA emergency. The most common operative services were orthopedic surgery (3778, 35.6%), neurological surgery (1350, 12.7%), and general surgery (1611, 15.2%). Median [IQR] booking and actual case durations were 210 (133, 242) min and 160 (93, 257) min, respectively. Clinical characteristics were well-balanced between the training and validation sets, with SMD < 0.1 for all variables (Table 1), including differences in case duration (Appendix A).

### 3.2. Preoperative and Intraoperative Variables

In the first variable-selection step, 14 preoperative and 20 intraoperative variables were evaluated by fitting an elastic net model on the training set (Appendix A). Eleven time-sensitive intraoperative variables were represented as the number of minutes below and above nine population-level distributional thresholds (Appendix A). With the addition of missingness indicator variables, this led to a total of 226 candidate variables. Preoperative variables retained by the model were the preoperative sCr closest to the procedure (last preoperative sCr within 90 days) and the minimum sCr within 90 days before the procedure. Retained intraoperative variables were the number of minutes the pulse oximetry perfusion index was ≤0.8%, minutes urine output ≤31.29 mL/h, minutes pulse oximetry heart rate > 85, and total estimated blood loss.

We then explored interactions among the retained main terms to capture the interplay among physiological processes leading to AKI. To determine interactions among the six variables identified above, a glinternet model was fit on the training set. Retained interactions were between the last preoperative sCr and perfusion index, urine output, and estimated blood loss; between the urine output and minimum preoperative sCr and perfusion index; between the minimum preoperative sCr and heart rate; and between the estimated blood loss and heart rate and perfusion index. Coefficients of the POSTOP-AKI model are presented in Appendix A. A web application that calculates the POSTOP-AKI score for user-specified inputs is available at https://postop-aki.onrender.com (accessed on 30 July 2023; Figure 2).

### 3.3. Predictive Performance of the POSTOP-AKI Model

We then evaluated the predictive performance of the POSTOP-AKI model in the validation set. The ROC AUC was 0.81 (95% CI, 0.77–0.85) for the POSTOP-AKI model and 0.75 (0.70–0.80) for the 90-day minimum preoperative sCr (Figure 3A). The PR AUC was 0.24 for the POSTOP-AKI model and 0.19 for the 90-day minimum preoperative sCr (Figure 3B). Decision curve analysis demonstrated an increased net benefit of the POSTOP-AKI model compared to a treat-all strategy, treat-none strategy, and prediction using the minimum preoperative sCr across a wide range of threshold probabilities (Figure 3C). The increased predictive performance of the POSTOP-AKI model was consistent across a range of cutoffs in the 90-day minimum preoperative sCr (Appendix A). Both the relationship between the predicted postoperative increase in sCr (the POSTOP-AKI score) and the observed postoperative increase in sCr and between the POSTOP-AKI score and the observed rate of AKI were largely linear (Appendix A). *R*^2^ and RMSE for the calibration curve in the validation set were 0.22 and 0.18, respectively. For comparison to the POSTOP-AKI model, the random forest model had an ROC AUC of 0.79 and a PR AUC of 0.22 and the XGBoost model had an ROC AUC of 0.80 and a PR AUC of 0.21 (Appendix A).

Low, middle, and high score thresholds were 0.05, 0.0767, and 0.1 for the POSTOP-AKI model, respectively, and 0.75, 0.945, and 1.25 for the 90-day minimum preoperative sCr, respectively (Table 2). For the low cutoff in the POSTOP-AKI model, specificity was 37%, sensitivity was 91%, and PPV was 8.2%. For the low cutoff in the minimum preoperative sCr, specificity was 47%, sensitivity was 83%, and PPV was 8.7%. For the middle cutoff in the POSTOP-AKI model, specificity was 77%, sensitivity was 74%, and PPV was 15%. For the middle cutoff in the minimum preoperative sCr, specificity was 72%, sensitivity was 67%, and PPV was 12%. For the high cutoff in the POSTOP-AKI model, specificity was 87%, sensitivity was 60%, and PPV was 19%. For the high cutoff in the minimum preoperative sCr, specificity was 88%, sensitivity was 47%, and PPV was 18%.

### 3.4. Associations of Model Variables with Postoperative AKI

To aid model interpretation, an OLS linear regression model was fit on the training set using the main terms retained in the POSTOP-AKI model (Appendix A). Variables retained in the POSTOP-AKI model demonstrated largely linear relationships with the observed rate of AKI, supporting the use of a linear model (Appendix A).

We also assessed the relationship between time-sensitive intraoperative variables retained in the POSTOP-AKI model and postoperative AKI at each minute throughout the procedure. From the procedure start time to up to 10 h intraoperatively, the proportion of cases meeting the predictive thresholds for urine output, perfusion index, and heart rate was consistently higher among those who developed postoperative AKI compared to those who did not (Figure 4). Throughout intraoperative time points, the median [IQR] difference in the portion meeting the predictive threshold between cases that developed AKI and those that did not was 15.8% (13.3–19.1%) for urine output, 8.0% (5.0–11.3%) for heart rate, and 7.2% (5.2–9.5%) for perfusion index.

Time-series distributions of urine output, perfusion index, and heart rate, as well as density plots of estimated blood loss and 90-day minimum preoperative sCr, are presented in Appendix A. Distributions were largely parallel, with urine output and perfusion index shifted lower in cases developing AKI, and heart rate, estimated blood loss, and 90-day minimum preoperative sCr shifted higher in cases developing AKI.

## 4. Discussion

Anesthesia is an ideal discipline for integrating data analysis and machine learning into clinical decision support because much of the raw clinical data were digitized and automatically captured and stored. We report a simple and easy to use prediction model for postoperative AKI that uses non-invasive intraoperative variables. Notable attributes of our model are its simplicity, interpretability, ease of intraoperative use, and potential actionability. Underlying this model, we developed a statistical method for transforming intraoperative time-series data into a set of numeric variables. This method may prove useful for other applications of prediction models in the operative setting.

Accurate intraoperative prediction of AKI could enable customizable reno-protective interventions for patients who will benefit from them most, such as hemodynamic optimization with vasoactive drugs and infusions of fluid and blood products [33], without exposing patients with a low probability of AKI to these interventions. The prediction can also improve the transfer of the information and concerns to those who take care of the patient after surgery, as it has been shown that important details are often lost when a handoff occurs between providers [34].

Our analysis identified a set of predictive intraoperative variables that matches clinical intuition. Urine output, heart rate, and estimated blood loss are all thought to reflect volume status and fluid responsiveness. Of note, the pulse oximetry perfusion index was retained in the model over more central measures of perfusion, such as blood pressure. The perfusion index is the ratio of absorbed arterial inflow light, or pulsatile pulse oximetry signal, to the nonpulsatile signal. While the perfusion index has been previously proposed as a marker of peripheral perfusion [35,36], its use for the prediction of postoperative AKI is new to our knowledge. The predictive ability of the perfusion index in our application suggests that renal hypoperfusion may not always coincide with central hypotension.

Of the intraoperative variables included in the POSTOP-AKI model, both heart rate and perfusion index are continuous, noninvasive, and routinely measured for almost all procedures. While formal documentation of the urine output and estimated blood loss is more intermittent in nature, particularly in cases not using a foley catheter for urine output, most anesthesiologists closely monitor these values throughout procedures.

The baseline kidney function, as reflected by preoperative sCr, is perhaps the most common and straightforward predictor of postoperative AKI widely available currently [37,38,39]. The Simple Postoperative AKI Risk (SPARK) model is a relatively simple score-based model that uses age, estimated glomerular filtration rate, sex, expected surgical duration, emergency operation, diabetes mellitus status, renin–angiotensin–aldosterone system blockade use, hypoalbuminemia, anemia, and hyponatremia [40]. Lastly, more complex predictive models of postoperative AKI have been reported using hundreds of preoperative and intraoperative variables [12]. In comparison to preoperative sCr, the POSTOP-AKI model demonstrated a clinically meaningful increase in predictive performance. The POSTOP-AKI model used fewer preoperative variables than the SPARK model and achieved superior predictive performance. Lastly, the POSTOP-AKI model achieved similar performance to complex preoperative and intraoperative models while including a far more limited set of variables and a more interpretable linear model.

In the POSTOP-AKI model, several intraoperative variables were retained over preoperative variables despite placing no constraints on inclusion or exclusion of either set of variables. While it may be possible to achieve a similar predictive performance with a much larger set of candidate preoperative variables and the use of a more complicated model, a strength of our model is its simplicity and ease of use. Conversely, even among larger, more complicated models, the inclusion of the intraoperative variables identified in this study or the use of our method for transforming intraoperative time series data may still lead to meaningful increases in predictive performance compared to entirely preoperative models. These questions may serve as the basis for future research.

Also subject to further research is the question of optimal score cutoffs for the POSTOP-AKI model. While we describe three potential cutoffs, a formal investigation of the risks and benefits of intervention based on different cutoffs is perhaps better served by a prospective or randomized study. Further work is needed to determine whether clinical interventions guided by the predicted risk of AKI can alter rates of postoperative AKI.

This study has some limitations. Our model was developed and validated at a single center, and broader application necessitates validation of the model with external data. Importantly, score thresholds for the binary outcome were intended to be demonstrative and are not optimized for external data. Nevertheless, our data represented a broad range of operative services, anesthesiologists, and surgeons, and the retained model variables have clear physiologic reasons to be predictors of AKI. Our study design limits a causal interpretation of the results, which strictly requires that exposures precede the outcome. While we attempted to exclude those with preoperative AKI, concurrent timing of the exposures and outcome is still possible given the lack of real-time indicators of AKI. Another important limitation of our study is the possibility that the accuracy of the perfusion index varies in different skin tones, as observed for the pulse oximetry oxygen saturation [41].

## 5. Conclusions

We report a simple and easy to use prediction model for postoperative AKI that uses preoperative sCr values, the number of minutes meeting cutoffs for urine output, heart rate, and perfusion index intraoperatively, and the total estimated blood loss. A web application that calculates the model score is available online. Our findings demonstrate the utility of intraoperative time-series data for prediction problems in the operative setting, including a new potential use of the perfusion index. Further research is needed to prospectively characterize the use of the POSTOP-AKI model in clinical settings.

## Figures and Tables

**Figure 1 bioengineering-10-00932-f001:**
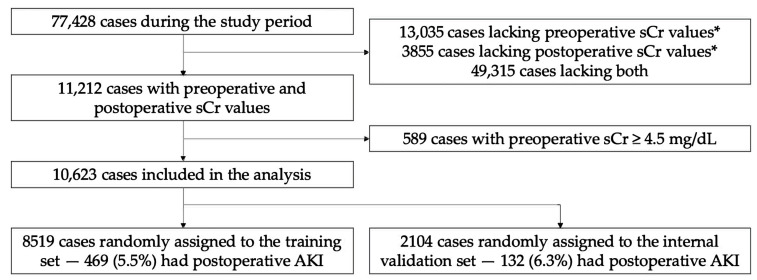
Cohort development for training and internal validation. * Within the time interval 90 days preceding surgery and 48 h following surgery. sCr, serum creatinine.

**Figure 2 bioengineering-10-00932-f002:**
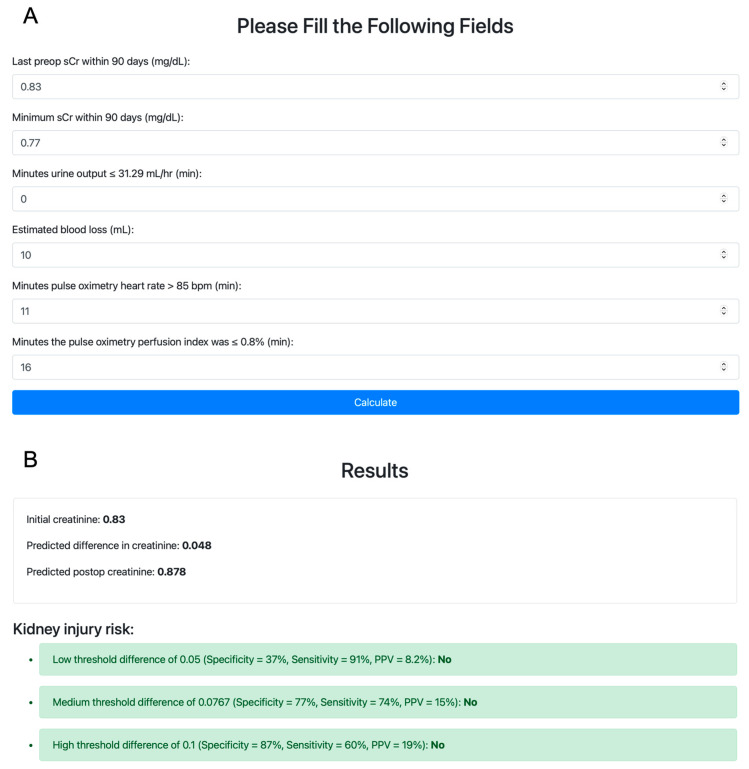
Demonstration of the POSTOP-AKI web application. (**A**) Variables are initialized with their respective medians in the training set. POSTOP-AKI, perfusion optimized score to predict AKI. (**B**) The predicted postoperative increase in serum creatinine is compared to three thresholds for predicting postoperative acute kidney injury.

**Figure 3 bioengineering-10-00932-f003:**
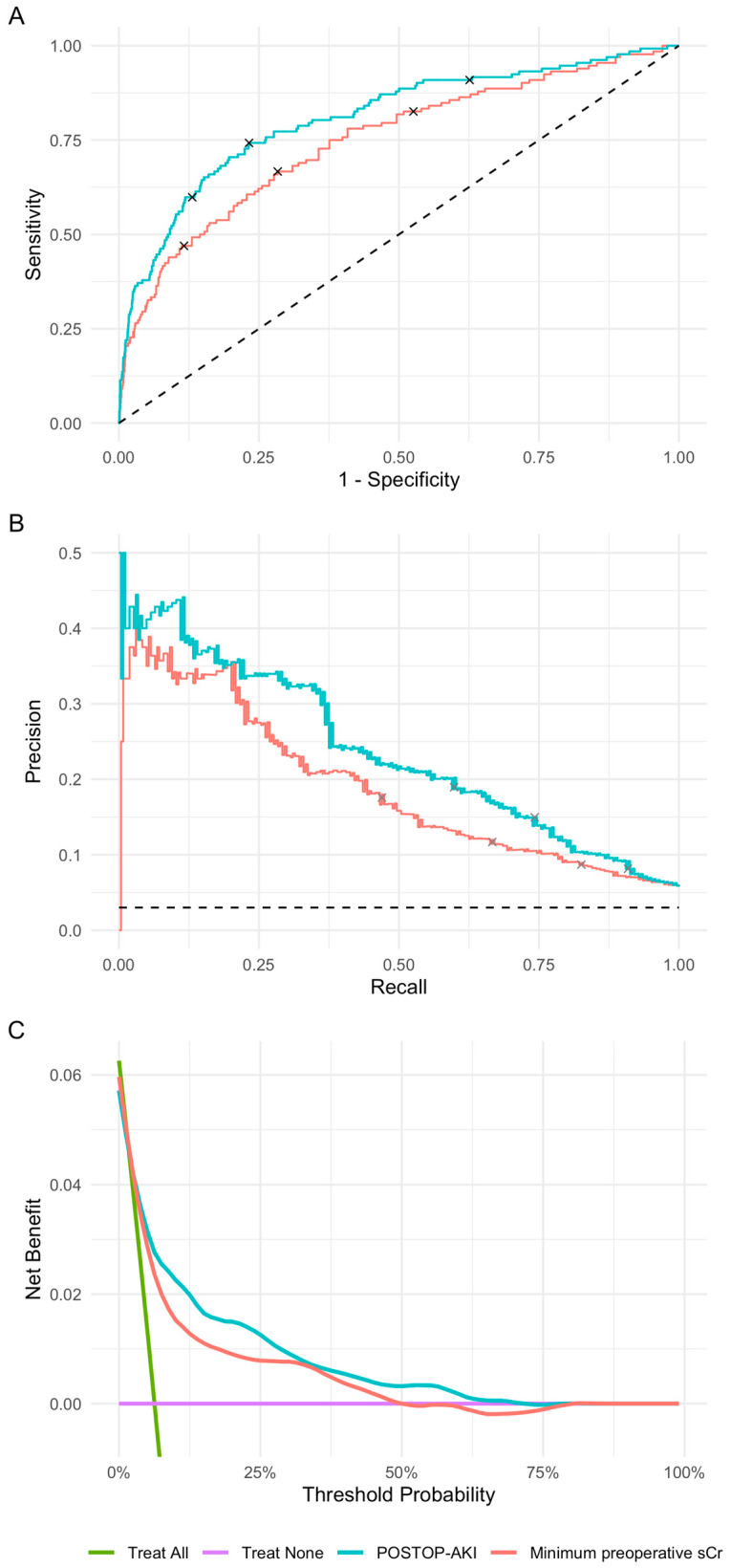
Performance metrics in the validation set. (**A**) Receiver operating characteristic curves. The 95% confidence intervals were calculated using the DeLong method. (**B**) Precision-recall curves. The dashed line represents a random classifier. Points indicated on the curves correspond to cutoffs demonstrated Table 2. (**C**) Decision curve analysis. POSTOP-AKI, perfusion optimized score to predict AKI.

**Figure 4 bioengineering-10-00932-f004:**
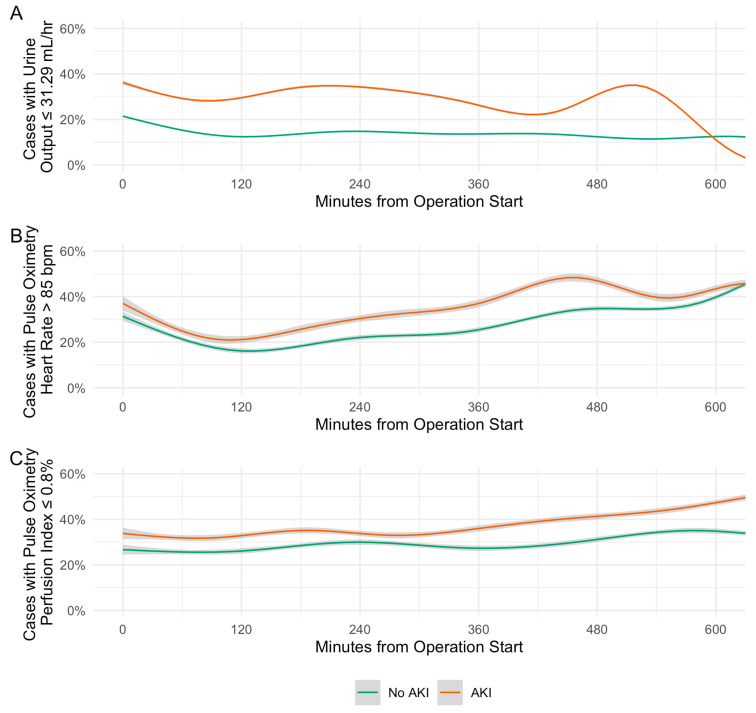
Distributions of intraoperative predictors of acute kidney injury. (**A**) Urine output. (**B**) Pulse oximetry heart rate. (**C**) Pulse oximetry perfusion index. Trend lines are generalized additive models. Grey shading represents 95% confidence interval.

**Table 1 bioengineering-10-00932-t001:** Characteristics of the study population. SMD, standardized mean difference. Data missingness (%) was 0 with the exception of 3.2, 2.8, and 4.7 for height, weight, and body mass index; 39 and 37 for preoperative temperature and heart rate; 37 and 36 for ASA class and emergency; 18 and 17 for booking case length and surgical risk score.

Characteristic	Overall	Training Set	Validation Set	SMD
N	10,623	8519	2104	
Age (median [IQR])	62 (51, 72)	62 (51, 72)	62 (50, 71)	0.02
Male (%)	5871 (55.3)	4747 (55.7)	1124 (53.4)	0.05
Race (%)				0.02
American Indian or Alaska Native	89 (0.8)	74 (0.9)	15 (0.7)	
Black or African American	916 (8.6)	732 (8.6)	184 (8.7)	
Other, including multiracial	3217 (30.3)	2573 (30.2)	644 (30.6)	
White or Caucasian	6401 (60.3)	5140 (60.3)	1261 (59.9)	
Hispanic or Latino (%)	1558 (14.7)	1246 (14.6)	312 (14.8)	0.006
Weight—kg (median [IQR])	77 (64, 91)	77 (64, 91)	77 (64, 91)	0.01
Height—cm (median [IQR])	170 (163, 178)	170 (163, 178)	168 (163, 178)	0.08
Body mass index—kg/m^2^ (median [IQR])	26 (22, 30)	26 (22, 30)	26 (22, 30)	0.03
Diabetes mellitus (%)	257 (2.4)	208 (2.4)	49 (2.3)	0.007
Hypertension (%)	296 (2.8)	247 (2.9)	49 (2.3)	0.04
Heart failure (%)	119 (1.1)	95 (1.1)	24 (1.1)	0.002
Liver disease (%)	401 (3.8)	327 (3.8)	74 (3.5)	0.02
Surgical risk score (%)				0.02
0	6554 (61.7)	5238 (61.5)	1316 (62.5)	
1	2571 (24.2)	2070 (24.3)	501 (23.8)	
2	1498 (14.1)	1211 (14.2)	287 (13.6)	
American Society of Anesthesiologists (ASA) class (%)				0.05
1	216 (2.0)	163 (1.9)	53 (2.5)	
2	5853 (55.1)	4685 (55.0)	1168 (55.5)	
3	3957 (37.2)	3184 (37.4)	773 (36.7)	
4	591 (5.6)	483 (5.7)	108 (5.1)	
5	6 (0.1)	4 (0.0)	2 (0.1)	
ASA emergency (%)	2675 (25.2)	2138 (25.1)	537 (25.5)	0.01
Primary service (%)				0.07
Cardiac surgery	188 (1.8)	151 (1.8)	37 (1.8)	
Cardiology	826 (7.8)	680 (8.0)	146 (6.9)	
Gastroenterology	108 (1.0)	90 (1.1)	18 (0.9)	
General surgery	1611 (15.2)	1277 (15.0)	334 (15.9)	
Neurological surgery	1350 (12.7)	1099 (12.9)	251 (11.9)	
Orthopedic surgery	3778 (35.6)	3003 (35.3)	775 (36.8)	
Other	1012 (9.5)	823 (9.7)	189 (9.0)	
Plastic surgery	324 (3.0)	257 (3.0)	67 (3.2)	
Thoracic surgery	240 (2.3)	193 (2.3)	47 (2.2)	
Vascular surgery	1186 (11.2)	946 (11.1)	240 (11.4)	
Booking case length—min (median [IQR])	210 (133, 242)	210 (134, 242)	210 (130, 241)	0.02
Actual case duration—min (median [IQR])	160 (93, 257)	161 (94, 257)	159 (90, 258.25)	0.003
Intraoperative				
Use of inhalational anesthetic (%)	9827 (92.5)	7878 (92.5)	1949 (92.6)	0.006
Use of pressors (%)	6468 (60.9)	5188 (60.9)	1280 (60.8)	0.001
Median temperature (median [IQR])	97 (96, 98)	97 (96, 98)	97 (97, 98)	0.04
Median heart rate (median [IQR])	72 (63, 83)	72 (63, 83)	72 (63, 82)	0.02
Median systolic blood pressure (median [IQR])	112 (103, 124)	112 (103, 124)	112 (103, 125)	0.01
Median respiratory rate (median [IQR])	12 (10, 14)	12 (10, 14)	12 (10, 14)	0.008

**Table 2 bioengineering-10-00932-t002:** Confusion matrices for low, medium, and high cutoffs of predicted postoperative increase in serum creatinine (sCr).

	Minimum Preoperative sCr		POSTOP-AKI	
Score threshold	<0.75	≥0.75	<0.05	≥0.05
No AKI	935	1037	737	1235
AKI	23	109	12	120
Score threshold	<0.945	≥0.945	<0.0767	≥0.0767
No AKI	1413	559	1514	458
AKI	44	88	34	98
Score threshold	<1.25	≥1.25	<0.1	≥0.1
No AKI	1743	229	1714	258
AKI	70	62	53	79

## Data Availability

The data presented in this study are available in a deidentified format via a secure data repository on request from the corresponding author.

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
