# Peer review of "Development of a Machine Learning Model of Postoperative Acute Kidney Injury Using Non-Invasive Time-Sensitive Intraoperative Predictors"

_bioengineering, 2023, doi:10.3390/bioengineering10080932_

Round 1

Reviewer 1 Report

This paper employs machine learning for the task of
predicting postoperative AKI using preoperative and intraoperative
variables . Experimental results demonstrate the feasibility
of machine learning in tackling the learning task. The paper i well
motivated. However, the following missing details need
to be mentioned as follows.

Major Comments:
- Include a demo (or example) in the website via initializing variable values used to calculate the outcome.

-It is stated in the abstract that "Fourteen preoperative and 20 intraoperative variables were evaluated by elastic net followed by hierarchical group least absolute shrinkage and selection operator (LASSO) regression".
it means that elastic net was used for variable selection followed by
LASSO for variable selection in which that is considered as a hierarchical feature selection approach?. Then, need to
specify what model was then used for training and performing prediction. Therefore, include a flowchart showing from the input to selecting variables
to predicted outcomes (and also include model names in each stage). Also, there is a need to justify the selection of the threshold for mapping to binary outcomes

-Need to include other machine learning algorithms  in the comparison such as Random Forest (using "randomForest" package in R), XGBOOST (using "xgboost" package in R), SVM (using "e1071" package in R).

Author Response

Reviewer #1

This paper employs machine learning for the task of predicting postoperative AKI using preoperative and intraoperative variables. Experimental results demonstrate the feasibility of machine learning in tackling the learning task. The paper is well motivated. However, the following missing details need to be mentioned as follows.

Major Comments:

- Include a demo (or example) in the website via initializing variable values used to calculate the outcome.

Thank you for your suggestion. Please see the new reference to Figure 2 on page 6, line 256. The following figures are renumbered accordingly.

-It is stated in the abstract that "Fourteen preoperative and 20 intraoperative variables were evaluated by elastic net followed by hierarchical group least absolute shrinkage and selection operator (LASSO) regression". It means that elastic net was used for variable selection followed by LASSO for variable selection in which that is considered as a hierarchical feature selection approach? Then, need to specify what model was then used for training and performing prediction. Therefore, include a flowchart showing from the input to selecting variables to predicted outcomes (and also include model names in each stage). Also, there is a need to justify the selection of the threshold for mapping to binary outcomes

- Thank you for your suggestion. Please see the new reference to Figure S1 on page 3, line 149. The following figures are renumbered accordingly.

- The thresholds selected for mapping to binary outcomes were intended to be demonstrative rather than definitive, which would require a formal cost-benefit analysis that is beyond the scope of this paper. Please see page 4, line 163 for discussion of the rationale for these thresholds. The demonstrative nature of the thresholds is also now discussed as a limitation on page 9, line 389.

-Need to include other machine learning algorithms  in the comparison such as Random Forest (using "randomForest" package in R), XGBOOST (using "xgboost" package in R), SVM (using "e1071" package in R).

- We appreciate the reviewer’s point about evaluating other models. The main objective of our paper was to develop a relatively simple and interpretable predictive model for postoperative AKI. For this reason, we did not report more complex models fit on all the candidate variables. For comparison, we do now report the results of random forest (we use the faster ranger implementation) and xgboost fit on the limited set of variables selected by elastic net. We found that these models did not improve prediction performance compared to the POSTOP-AKI model. Moreover, the POSTOP-AKI model is relatively simple to implement in real time by applying its coefficients in a web application. On the other hand, more complex tree-based models require integration into the electronic medical record.

- Please add additions at page 4, line 183, and page 6, line 273.

- The authors thank the reviewer for their careful review and helpful comments, which have greatly improved the paper.

Reviewer 2 Report

The paper presents interesting findings and is a candidate for publication. However, before it can be accepted, several major concerns must be addressed:

  1. The authors claim to have followed the TRIPOD & STROBE guideline checklist, but these documents are not included in the supplementary material.

  2. A preamble explaining the process in lines 100-105 (referring to Table S2) would improve understanding for a wider audience.

  3. Figures are mentioned in the text, but none are included in the main manuscript.

  4. Figure S3 shows a calibration curve for the validation set. Could the authors provide more explanation? The curve suggests that the model may not predict well. Please report R2 and RMSE.

  5. Supplementary tables S3 and S4 contain too many decimal places. Please report with the appropriate number of significant figures.

  6. What is the rationale for testing interactions? Is there a clinical or physiological reason? This increases model complexity and may require the use of marginal effects to better understand variable impact.

  7. Lines 254-256 state that the model demonstrated largely linear relationships. Why is this problematic? Splines can be used with OLS to model non-linear effects and potentially improve predictive ability. Can the authors comment on this and add their rationale to the manuscript?

  8. The handling of time-series data is unclear. Would a mixed-effect model be more appropriate for dealing with autocorrelation of variables? (optional)

  9. The data availability statement mentions patient confidentiality, but data could be anonymized and stored in a public repository such as Zenodo.

Author Response

Reviewer #2

The paper presents interesting findings and is a candidate for publication. However, before it can be accepted, several major concerns must be addressed:

  1. The authors claim to have followed the TRIPOD & STROBE guideline checklist, but these documents are not included in the supplementary material.

Thank you for pointing out this omission. The checklists are now included in the supplementary material.

  1. A preamble explaining the process in lines 100-105 (referring to Table S2) would improve understanding for a wider audience.

Thank you for your suggestion. Please see page 3, line 100 for clarification.

  1. Figures are mentioned in the text, but none are included in the main manuscript.

Thank you for bringing this to our attention. We included in-line figures in our submission, and it appears those were removed during the editorial process. We will be sure to upload the figures separately for our revision.

  1. Figure S3 shows a calibration curve for the validation set. Could the authors provide more explanation? The curve suggests that the model may not predict well. Please report R2 and RMSE.
  • We thank the reviewer for raising this important point. We chose a linear regression model converted to a binary classification not necessarily to predict the postoperative sCr but because we wanted to provide the model with more information on gradations in postoperative sCr increase in the setting of physiological insults leading to AKI. Based on KDIGO criteria, which we use clinically at our institution, AKI is a binary outcome. Therefore, we focused our paper on predictive metrics for the binary outcome. Our review of the literature revealed that prior models for predicting AKI directly modeled a binary outcome without taking advantage of the underlying continuous nature of sCr. We therefore feel that our approach is a strength that builds on prior work.
  • Please see page 6, line 272 for R2 and RMSE of the calibration curve in the validation set, which we feel are acceptable insofar as they facilitate accurate binary predictions.
  1. Supplementary tables S3 and S4 contain too many decimal places. Please report with the appropriate number of significant figures.
  • Thank you for bringing this oversight to our attention. Table S3 and S4 now report 4 and 2 significant figures, respectively. We report coefficients in Table S3 with more precision because they are the POSTOP-AKI model coefficients, which readers may be interested in using to make predictions.
  1. What is the rationale for testing interactions? Is there a clinical or physiological reason? This increases model complexity and may require the use of marginal effects to better understand variable impact.
  • We appreciate the reviewer’s point and acknowledge that adding interaction terms increased the complexity of our model. There was a clinical motivation for doing so. The physiological events leading to AKI involve a complex interplay among organ systems. In this context, it is unlikely that individual physiological effects are independent, as would be assumed by a model with only main terms. To give one example, the interaction term between urine output and perfusion index may reflect the interaction of peripheral hypoperfusion and decreased renal function. While the addition of interaction terms increases model complexity, the user inputs only the main terms in the web application. Hence the model with interaction terms is still easier to implement than a tree-based model or neural net, which would require integration into the electronic medical record.
  • Please see page 6, line 247 for clarification on the rationale for testing interaction terms.
  1. Lines 254-256 state that the model demonstrated largely linear relationships. Why is this problematic? Splines can be used with OLS to model non-linear effects and potentially improve predictive ability. Can the authors comment on this and add their rationale to the manuscript?
  • Thank you for asking for clarification. We assessed the linearity of variable associations precisely to determine whether the use of non-linear methods, such as splines, might be warranted. Since we found that univariate associations were largely linear, we did not pursue a nonlinear model. While a nonlinear model may improve predictive ability, its added complexity is a detriment to interpretability. We now compare the POSTOP-AKI model to tree-based models, which are highly nonlinear, and report no benefit in predictive ability.
  • Please see our clarification on page 8, line 304.
  1. The handling of time-series data is unclear. Would a mixed-effect model be more appropriate for dealing with autocorrelation of variables? (optional)
  • We appreciate this helpful suggestion. We agree that mixed-effect models are particularly helpful for repeated measures data, such as time series. The reviewer’s point about autocorrelation is particularly salient when it comes to statistical inference using repeated measures. Because our paper was more focused on prediction rather than inference, we opted to transform our time series data into a time-invariant format prior to model building. This allowed us to use simpler models that are easier to implement and interpret. Analysis of our time series data with a mixed-effects model is certainly an interesting future direction.
  1. The data availability statement mentions patient confidentiality, but data could be anonymized and stored in a public repository such as Zenodo.

- Thank you for the suggestion. We are happy to provide our anonymized dataset, pending necessary updates to our IRB. Institutional policy prohibits us from uploading patient-level data, even anonymized, on an open repository such as Zenodo. We will instead use Vivli, a controlled-access clinical data repository.

- Please see page 10, line 450 for an updated data availability statement. We anticipate it will take a few more days to complete the necessary forms and upload the data. We will be sure to complete these tasks by time of publication.

- The authors thank the reviewer for their careful review and helpful comments, which have greatly improved the paper.

Round 2

Reviewer 1 Report

Authors addressed raised comments in a previous round of review

Author Response

Authors addressed raised comments in a previous round of review

Thank you for your review of this manuscript.

Reviewer 2 Report

Just a couple of remarks (minor).

Regarding Figure S4 (revised manuscript), I can appreciate the author's rationale, but I remain unconvinced by the current presentation of the calibration curve. The scatter plot makes it difficult to grasp the predictive model's utility. To improve clarity, I suggest considering a LOESS (Locally Weighted Scatterplot Smoothing) curve fitting technique. This would provide a smoother curve that better illustrates the predictive performance of the model. Or, just remove from the manuscript.

Additionally, while the STROBE statement does include the corresponding lines, it lacks the relevant text from the manuscript itself. Simply copying and pasting the information may not be sufficient. 

Author Response

Just a couple of remarks (minor).

Regarding Figure S4 (revised manuscript), I can appreciate the author's rationale, but I remain unconvinced by the current presentation of the calibration curve. The scatter plot makes it difficult to grasp the predictive model's utility. To improve clarity, I suggest considering a LOESS (Locally Weighted Scatterplot Smoothing) curve fitting technique. This would provide a smoother curve that better illustrates the predictive performance of the model. Or, just remove from the manuscript.

  • Thank you for the suggestion. Figure S4 now includes a smooth fit using LOESS regression. In addition to the LOESS trend, we kept the linear trend line, as this is the line referenced by our reported R2 and RMSE. To help readers appreciate these two lines, we made the points on the scatterplot smaller and more transparent.

Additionally, while the STROBE statement does include the corresponding lines, it lacks the relevant text from the manuscript itself. Simply copying and pasting the information may not be sufficient

  • Thank you for the suggestion. The STROBE statement now includes relevant text from the manuscript where applicable.

Thank you for your review of this manuscript.